# Long-term shareholder perks and stock price reaction

**Yao Gao**[1], **Yoshiaki Nose**[2]*

**1** Doshisha Business School Network, Kyoto, Japan, **2** Graduate School of Business, Doshisha University, Kyoto, Japan

* ynose@mail.doshisha.ac.jp

## Abstract

This study aimed to examine the impact and source of announcements introducing additional long-term shareholder perks on stock prices of Japanese listed companies. We produced more precise analysis results by categorizing the total sample into favorable change and unfavorable change sample. As a result, we found that long-term shareholder perks have a positive impact on stock prices through the expansion of the shareholder base in the case of a favorable change, whereas there is no negative impact on stock liquidity due to an increased number of long term individual shareholders. On the contrary, in the case of an unfavorable change, we found a weak trend of shrinkage in the shareholder base due to individual shareholders' defection and a consequent decrease in stock liquidity. In the case of a favorable change, the long-term shareholder perks program functions as a means to increase the number of shareholders and encourage them to hold the shares for a longer period of time.

**Data Availability Statement:** All relevant data are within the paper and its Supporting Information files.

**Funding:** Yoshiaki Nose thanks Japan Society for the Promotion of Science (JSPS) Grants-in-Aid for Scientific Research (22K01566) for financial support. The funder had no role in study design,

## Introduction

This study analyzes the impact of the additional introduction of a long-term shareholder perks program on the stock prices of Japanese companies that have implemented such programs, also identifying the source of the impact. The analysis period covers 11 years, from 2010 to 2021. In Japan, shareholder perks, which provide shareholders with the company's goods or services in addition to dividends, have been established as a form of payout. In recent years, it has become clear that shareholder perks offer benefits that differ from dividends and share buybacks [1–3], and interest in shareholder perks is also growing outside Japan [4].

In Japan, where shareholder perks are well developed, a system has been introduced to favor long-term shareholders. Such perks are provided to shareholders who hold shares beyond a certain period of time. The primary purpose of a company's introduction of a preferential long-term shareholder perks program is to encourage individual shareholders to hold shares for a long period of time [5]; however, the impact of long-term shareholder programs on companies' stock performance remains unclear. It is also not clear whether such programs actually encourage individual shareholders to hold shares for the long term.

data collection and analysis, decision to publish, or preparation of the manuscript.

**Competing interests:** The authors declare no conflict of interest.

The purpose of this study is twofold. The first intention is to determine the effect of announcements of the introduction of long-term shareholder perks on the stock price of firms that implement these programs. The second is to obtain some insight into the source of the identified stock price effect. Previous studies have found that the announcement of the first shareholder perks program has a positive effect on stock prices (e.g., [2]); however, the impact of the introduction of "additional long-term" shareholder perks programs is unclear. This study is the first to examine the stock price effect of long-term shareholder perks.

We examine two types of long-term shareholder perks, those that do not change the amount of benefits offered to short-term shareholder perks (a favorable change for long-term shareholders), and those that reduce benefits for short-term shareholders (an unfavorable change). Although individual investors' evaluations of both types of changes may differ, a previous study has not distinguished between favorable and unfavorable changes [6]. By reporting the results of an analysis dichotomized into favorable and unfavorable changes, this study provides more accurate information on the impact of the introduction of long-term shareholder perks.

The following three notable points are demonstrated in this study. First, announcements of the introduction of a long-term shareholder perks program produce positive stock returns in the case of a favorable change. Second, a favorable change in long-term shareholder perks can cause a significant rise in the ratio of individual shareholders, whereas an unfavorable change decreases the ratio of individual shareholders. Furthermore, there is a significant positive correlation between stock returns and the increase in the ratio of individual shareholders. Third, we observed a trend toward higher stock liquidity from a favorable change, and conversely, a trend toward lower stock liquidity from an unfavorable change.

We find that long-term shareholder perks have a positive impact on stock prices through the expansion of the shareholder base in the case of a favorable change, while there is no negative impact on stock liquidity due to an increased number of long term individual shareholders. Conversely, in the case of an unfavorable change, we find a weak trend of shrinkage in the shareholder base due to individual shareholders' defection and a consequent decrease in stock liquidity. In the case of a favorable change, the long-term shareholder perks program functions as a means to increase the number of shareholders and encourage them to hold the shares for a longer period of time.

The structure of this paper is as follows. In Section 2, we outline the individual investor and investment environment in Japan, the shareholder perks system and the long-term shareholder perks system, and applicable definitions in this study. Section 3 surveys the previous studies in this research and presents our hypotheses. Section 4 describes the data and methods used in the empirical analysis. Section 5 presents the results of the analysis, and Section 6 summarizes the study.

## Shareholder perks

**Current status of shareholder perks.** Shareholder perks are payout programs in which a company offers its own goods, services, or vouchers as special gifts to shareholders who meet certain holding period conditions [7]. Because they are often not proportional to the number of shares held, shareholder perks are considered to be attractive to small shareholders [8,9]. For companies, shareholder perks are among the most representative measures for individual shareholder benefits, along with trading unit reductions and individual investor relations. Fig 1 presents the long-term shareholder perks program offered by Nissin Foods Holdings Co. The popular Cup Noodles and other products are sent once a year to all shareholders who hold at least 100 shares, which is the minimum trading unit. Shareholders who hold 300 or more shares for three years receive such products twice a year.

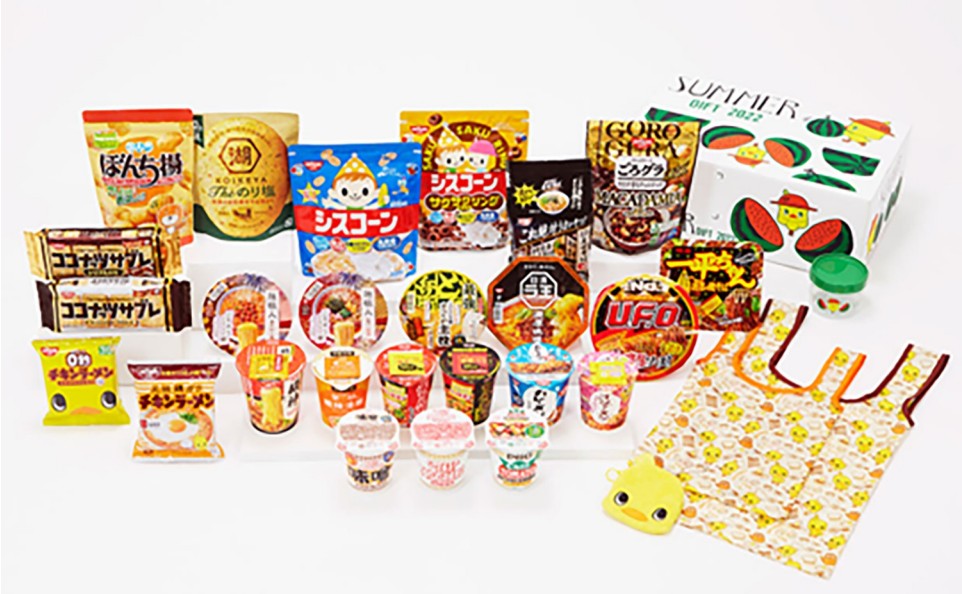

**Fig 1. Example of shareholder perks (Nissin Foods Holdings Co.).** Source: https://www.nissin.com/jp/ir/investors/preferential/.

The number of companies offering shareholder perks has increased in recent years. In 1993, about 300 companies offered such perks, but as of September 2022, 1,463 companies offered them. This represents about 40% of all listed companies [10]. According to a survey conducted by Nikkei, the most common answer regarding the purpose of implementing shareholder perks was "to encourage shareholders to hold shares for a long time." Shareholder perks programs have been criticized by institutional investors that cannot use the perks, contending that the cost of maintaining and managing perks is inflated (Nihon Keizai Shimbun, April 5, 2020, morning edition).

**Long-term shareholder perks programs.** As noted, long-term shareholder perks programs refer to a program that provides upgraded benefits to shareholders who hold their shares for a certain period of time (often more than one year). In recent years, an increasing number of companies have introduced long-term shareholder perks. As of March 31, 2020, 580 companies, or nearly 40% of the total number of companies offering shareholder perks, had introduced long-term shareholder perks program (Fig 2).

Long-term shareholder perks can be divided into two patterns when considering individual perks programs. The first pattern is that the benefits remain the same, with no required holding period, but new perks are added when a shareholder holds shares for a specified period of time. This is a change that is more attractive to new individual investors (favorable change). The second is the case in which long-term shareholder perks are added, while those without a required holding period are downgraded. In this case, perks that were originally available for a single day of investment are no longer available unless the shareholder invests for more than the specified time. In the second case, the shareholder perks that a new individual shareholder receives in the first year are reduced or eliminated, which is considered to be a change that is less attractive to new individual investors (unfavorable change). This indicates that the long-term shareholder perks programs do not simply grant positive benefits. Even if the amount of benefits increases, if the number of shares required and the required investment period also increase, individual investors will evaluate the perks negatively. Our study is unique in that it analyzes these two forms separately.

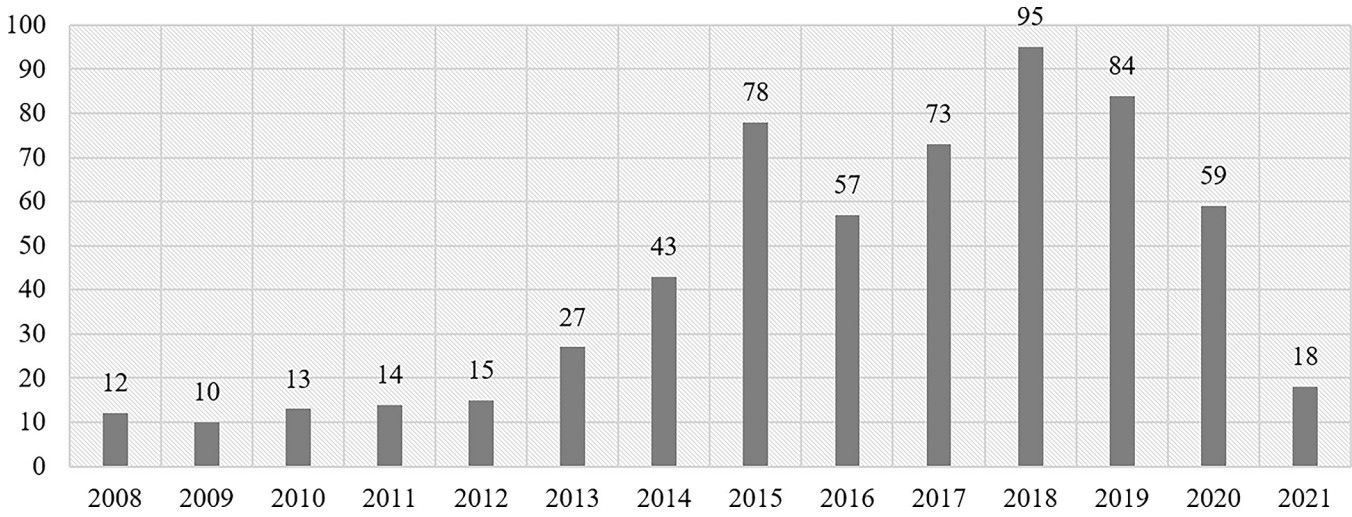

**Fig 2. Number of companies that have initiated long-term shareholder perks programs.** Note: Figures for 2021 are for the six-month period ending October 31, 2021. Source: Prepared by the authors with reference to each company's press release information.

Table 1 shows specific examples of favorable and unfavorable changes. In company (1), in addition to the traditional shareholder perks, long-term shareholders receive QUO Cards (a well-known money certificate in Japan), which is a favorable change. In company (2), after the change, the amount of QUO Cards granted to shareholders who have held their shares for more than three years is doubled, which is also a favorable change. In contrast, under company (3), shareholders who have not held shares for more than one year are not eligible for the special benefit, and in case (4), although there is a short-term perk, the amount of the perk is reduced, which are unfavorable changes.

**Table 1. Examples of favorable and unfavorable changes.**

| Favorable or unfavorable | case | Before change | After change | Summary |
|---|---|---|---|---|
| Favorable | 1 | 100 shares or more | In addition to the above | Another perks added to long term shareholders |
| | | Dining coupons worth 4,000 yen | 100 shares or more | |
| | | | 500 yen QUO card for 1 to 2 years of ownership | |
| | | | 1,000 yen QUO card for shares held for 2 years or more | |
| | 2 | 100 shares or more | 100 shares or more | Increased perks for long-term shareholders |
| | | 500 yen QUO card | 500 yen QUO card for less than 3 years | |
| | | | 1,000 yen QUO card for more than 3 years | |
| Unfavorable | 3 | 100 shares or more | 100 shares or more | Perks for long-term shareholders only |
| | | 500 yen QUO card | 1,000 yen QUO card for more than 1 year | |
| | 4 | 100 shares or more | 100 shares or more | Decrease in the amount of perks for short-term shareholders |
| | | 5000 yen QUO card | 2,000 yen QUO card for less than 1 year | |
| | | | 5,000 yen QUO card for more than 1 year | |

Source: Authors' construction based on companies' timely disclosure documents.

## Literature review

### Motivation for implementing shareholder perks

According to a survey by Daiwa Investor Relations (2022), companies implement shareholder perks for such purposes as long-term shareholder retention and to increase the number of individual shareholders [10]. Nose (2014) surveyed previous studies and categorized the motives for implementing shareholder perks programs into shareholder-related motives, stock performance-related motives, and other motivations. For shareholder-related motives, the author identified an increase in the number of shareholders, lengthened shareholding periods, dissolution of cross-shareholdings, prevention of hostile takeovers, and alternative means of dividend payments [11]. Regarding motives related to stock performance, the author cites increasing stock liquidity, destabilizing/stabilizing stock prices, and linking to stock indices. Other motivations include promoting the company's products and tax savings.

Yasutake and Nagata (2018) conducted a questionnaire-based survey of 3,702 listed companies in 2016 regarding the objectives of long-term perks implementation on the corporate side, determining that the two most significant objectives included increasing the number of individual shareholders and promoting long-term shareholding [12].

### Performance of companies implementing shareholder perks

Although the MM theorem implied that the presence or absence of shareholder perks should have no effect on stock prices [13], empirical data have demonstrated a positive stock price effect from their implementation. Karpoff et al. (2021) conducted an event study examining a sample of 307 firms that announced the introduction of shareholder perks between 2001 and 2011, finding the announcement of shareholder perks to have a positive short-term stock price effect [2]. The authors conclude that investors anticipate an increase in the number of shareholders and stock liquidity in companies that introduce shareholder perks, which leads to higher stock prices. Karpoff et al. (2021) also examined the impact of the abolition of shareholder perks, demonstrating that the cumulative abnormal return in the three days before and after the announcement date for the 81 firms that announced the abolition of shareholder perks was −5.89%, which was significantly negative at the 1% level [2]. A significantly negative stock price effect was also observed for firms that announced a dividend increase at the same time as the abolition of shareholder perks [14]. Considering that the announcement of the introduction of shareholder perks program was shown to have a positive stock price effect, while an announcement of the abolition of shareholder perks, even if accompanied by a dividend increase, had a negative stock price effect, it can be assumed that shareholder perks exert some positive added value. These results are consistent with Serita (2017), who found that shareholder perks have a positive effect on stock prices by lowering the implementing firm's cost of capital [15], and Uchida et al. (2022), who reported the share price support effect of shareholder perks during the 2008–2009 financial crisis [16].

Research has also been conducted regarding the long-term performance of stocks with shareholder perks. According to Nose et al. (2021), stock performance rises as the vesting date approaches for companies that offer shareholder perks [3]. According to the authors' financial visibility hypothesis, shareholder perks are introduced as a way to promote firms' presence to individual investors for stocks with low financial visibility among institutional investors.

While several studies have been conducted regarding shareholder perks programs, a limited number have examined long-term shareholder perks programs. Yasutake and Nagata (2018) analyzed the number of individual shareholders and Amihud's (2002) illiquidity index for all listed Japanese stocks from 2010 to 2016 [6,17]. The results demonstrated that long-term

shareholder perks in general significantly increased the number of individual shareholders at the 1% level, while significantly increasing illiquidity at the 5% level. This indicates that while shareholder perks are effective in increasing the number of shareholders, they reduce the liquidity of shares when long-term shareholder perks are favored, which contradicts past research.

## Implications and issues of previous studies

Although the motives and objectives for introducing shareholder perks may differ among firms, the main reasons for introducing them are to increase the number of individual shareholders and to encourage shareholders to hold shares for a long period of time. Correspondingly, many empirical studies have found the introduction of new shareholder perks to have a positive impact on stock prices by significantly increasing the number of shareholders and improving stock liquidity. Furthermore, although companies have introduced long-term shareholder perks to increase the number of long-term shareholders, the impact of the introduction of such perks on stock prices and stock liquidity remains unclear.

Specifically, previous studies have identified the following three issues. First, the short-term impact of the introduction of long-term shareholder perks on stock prices remains unclear. Second, the impact of the introduction of long-term shareholder perks on stock liquidity has not been determined. Third, when long-term shareholder perks are examined closely from the perspective of individual investors, as established above, there are both favorable and unfavorable changes, but previous studies have not distinguished between the two.

Our study presents useful empirical results on these issues by examining the effects of the introduction of long-term shareholder perks on stock prices and liquidity, and identifying the sources of perks' effects on stock returns.

## Hypotheses and analysis methods

### Hypotheses

This study references Suzuki and Isagawa's (2008) liquidity enhancement hypothesis, which assumes that the introduction of long-term shareholder perks will affect stock prices through the same mechanism as the introduction of new shareholder perks [18]. The liquidity hypothesis was proposed by Amihud and Mendelson (1986), who argued that financial policies that increase liquidity by increasing the number of shareholders will lower the cost of capital and raise shareholder value [19]. Merton's (1987), assertion that an expanding shareholder base reduces information asymmetry and positively affects stock prices is an additional theoretical base [20]. We assume that the introduction of additional "favorable" long-term shareholder perks for individual investors further attracts investors' attention and positively affects stock prices through an increase in the number of shareholders (expansion of the shareholder base), improved liquidity, and lower transaction costs; thus, the first hypothesis is as follows.

**H1 Announcement of a favorable long-term shareholder perk program generates a positive stock return**.

The introduction of a favorable long-term shareholder perks program produces the same effects as the introduction of a new shareholder perks program, aligning with the liquidity enhancement hypothesis. In contrast, an unfavorable long-term shareholder perk program either reduces or eliminates the perks for new individual shareholders in the first year. Therefore, we assume that only favorable changes are perceived as attractive by new shareholders and positively impact stock prices.

When a stock return is observed, interest shifts to the question of its source, and this study then tests the following two hypotheses.

**H2-1 Stock return is positively correlated with an increase in the ratio of individual shareholders**.

According to Merton's (1987) investor recognition hypothesis, an increase in the shareholder base leads to participation in trading by investors with diverse opinions regarding the future of the company, and investor diversification has a positive impact on the stock price [20]. Therefore, we assume that there will be a significant positive correlation between the stock price effect and the increase in the ratio of individual shareholders (shareholder base).

**H2-2 Stock return is positively correlated with increased stock liquidity**.

The hypothesis of liquidity increase contends that an expanding shareholder base has a positive effect on stock prices through an increase in stock liquidity, while Yasutake and Nagata (2022) asserted that long-term shareholder perks have a negative effect on stock liquidity [6]. In this study, we reference Suzuki and Isagawa (2008) to set our hypothesis [18].

This study analyzes all listed companies that announced the introduction of long-term shareholder perks during the 11-year period from April 2010 to the end of March 2021. First, we checked each firm's timely disclosure information and identified 308 firms. We next checked whether any other announcement was made that would affect the share price, such as increased or decreased profit, increased or decreased dividends, reduction in trading units, stock split or merger, or stock repurchase on the same day as the announcement of long-term shareholder perks. We found that 104 firms had same-day events, and our sample size is 203 firms after excluding these firms. The sample was further classified according to the method described in the previous section, identifying 128 firms with favorable changes and 75 firms with unfavorable changes.

Financial data for the sampled firms are obtained from firms' annual securities reports. Stock price data are collected from Bloomberg's stock price database.

## Stock price effect of announcements

To clarify the effect of long-term shareholder perks announcements on stock prices, we conduct an event study using the date of the announcement as the event date, referencing Karpoff et al. (2021) and Huang et al. (2022) [1,2].

First, we construct the following market model and estimate $\alpha i$ and $\beta i$ using an estimation period of 130 days from 150 days to 21 days before the event date.

$$R_{it} = \alpha_i + \beta_i R_{mt} + \varepsilon_{it} \tag{1}$$

where $R_{it}$ indicates the daily return of sample company $i$ on day $t$, $R_{mt}$ denotes the daily return of TOPIX on day $t$, and $\varepsilon_{it}$ is the error term.

Next, we estimate the daily abnormal return ($AR_i$) of sample firm $i$ from the above market model. AR is obtained by calculating the difference between the theoretical and actual returns for 21 days before and after the event date. The cumulative AR is the cumulative abnormal return (CAR). The significance of the mean and median of the AR and CAR for each day is tested using a t-test and a Wilcoxon rank sum test with AR = 0 as the null hypothesis.

## Changes in shareholder and stock liquidity

The changes in the number of shareholders and stock liquidity before and after the introduction of long-term shareholder perks are examined applying a difference-in-differences (DID)

analysis, which is conducted by setting control firms against sample firms, referencing Karpoff et al. (2021) and Huang et al. (2022) [1,2].

The control firms are identified using propensity score matching (PSM). Specifically, the group of firms that have introduced long-term perks programs is the sample (treatment) group, and firms that do not have a long-term perks program (but have a shareholder perks program) are the potential control group. The probability that these firms will introduce long-term shareholder perks (propensity score) is then calculated, and firms with similar characteristics to the sample firms are selected from the control group and matched on a one-to-one basis. Matching is conducted by aligning the timing over the calendar year. Because differences in industry can have as large an impact as financial and stock price variables, we ensured that the pairs had the same 17 TSE industry sectors. Eq (2) is the logit model used to calculate the propensity score.

$$
\begin{aligned}
\text{Prob}(&\text{dummy for introduction of long term shareholder peks}_{it} = 1) \\
&= F(\beta_0 + \beta_1 \ln sales_{it-1} + \beta_2 ROA_{it-1} + \beta_3 \ equity\ ratio_{it-1} + \beta_4 \ dividend\ yield_{it-1} \\
&\quad + \beta_5 \ volume\ turnover\ ratio_{it-1} + \beta_6 \ individual\ shareholders\ ratio_{it-1} \\
&\quad + \beta_7 \ indibilual\ shareholding\ ratio_{it-1})
\end{aligned}
$$

(2)

where $F$ is a logistic function, $i$ is the firm, and $t-1$ is the period immediately before the introduction of the long-term shareholder perks. The explained variable in the logit model is a dummy variable that equals 1 for firms that introduce long-term shareholder perks and 0 for firms that do not introduce it (but have shareholder perks with no required period). The financial data for each firm are obtained from firms' annual securities reports. The stock price data for each company are obtained from the Bloomberg database.

Table 2 presents the results of the balance evaluation after matching. t-tests are performed to verify whether statistically significant differences in the means of the covariate between the sample and control groups. The results show no statistically significant differences in any of the covariates.

Changes in the ratio of the number of individual shareholders and the ratio of individual shareholdings are examined using a DID analysis. We first calculate the amount of change (difference) in each variable from the period immediately prior to the date of the announcement of the introduction of long-term shareholder perks (−1 period) to the period immediately following the announcement (+1 period). Next, the difference between the difference for the sample firms and control firms is determined. Fig 3 illustrates the DID analysis.

The period when the announcement was made (period 0) is not included because of its short duration. The change for the sample firms alone is called the unadjusted difference, and the difference with control firms is called the adjusted difference.

**Table 2. Covariate checks.**

|  | Sample | Control | t-value |
|---|---|---|---|
| ln Sales | 10.97 | 10.8 | 1.07 |
| ROA | 6.95 | 6.33 | 0.98 |
| Equity Ratio | 47.04 | 48.81 | −0.83 |
| Dividend yield | 1.82 | 1.74 | 0.69 |
| Volume Turnover | 144.77 | 128.32 | 0.56 |
| Individual Shareholder Ratio | 96.44 | 96.13 | 0.92 |
| Individual Shareholding Ratio | 39.91 | 41.29 | −0.67 |

Individual shareholder ratio
Individual shareholding ratio
etc.

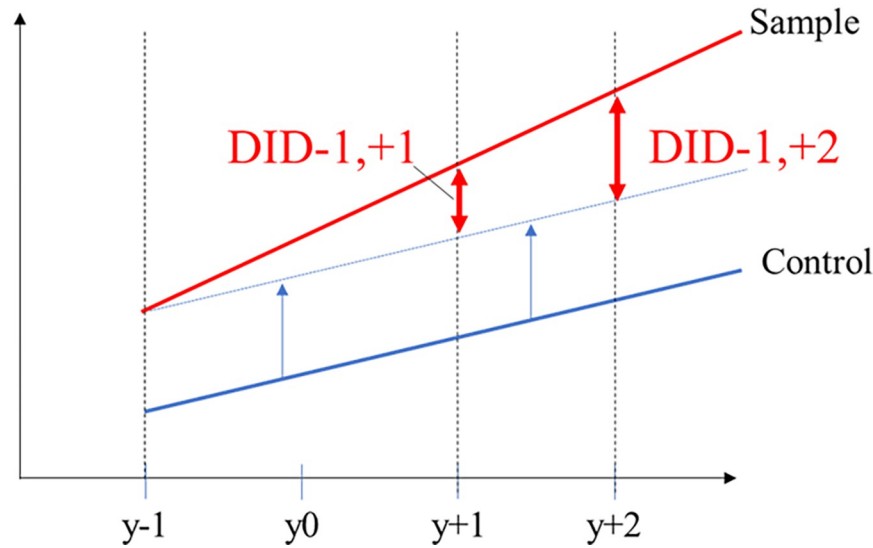

**Fig 3. Image of difference-in-differences analysis.**

We conduct a t-test for the mean and a Wilcoxon signed rank test for the median for these variables with the null hypothesis set to 0.

$$
\begin{aligned}
&\text{Unadjusted difference} \\
&= \text{Sample firms indicators[x period]} - \text{Sample firms indicators}[-1\text{ period}]
\end{aligned}
\tag{3}
$$

$$
\text{Adjusted difference} = \text{Change in sample firms} - \text{Change in control firms}
\tag{4}
$$

Our analysis of stock liquidity references Karpoff et al. (2021) and Huang et al. (2022) [1,2]. Specifically, a DID analysis is conducted to determine the quote spread ratio, the execution spread ratio, and Amihud's (2002) illiquidity indicator [17]. The quote spread ratio is the difference between the highest bid and lowest asking prices in the board information divided by the median of the two (Eq (5)), in which a higher value indicates a larger spread between buyers' and sellers' limit prices, and the more difficult it is to close a deal (i.e., low liquidity). Conversely, a smaller value indicates a closer spread between buyers' and sellers' limit prices, which makes it easier to close a deal (i.e., high liquidity). We reference Suzuki and Isagawa (2008), using the last quote of each trading day [18].

The execution spread ratio is the absolute value of the difference between the last execution price of a given day and the median quote price immediately following divided by the median quote price (Eq (6)). This is "the spread ratio paid by the trader of a stock in return for liquidity, and represents the size of the transaction cost" [18]. The execution spread ratio can also be interpreted as the larger (smaller) the value is, the lower (higher) the liquidity will be.

$$
\text{Quote spread ratio} = \frac{(\text{asking price} - \text{bid price})}{\text{median of both quotes}}
\tag{5}
$$

$$\text{Execution spread ratio} = \frac{|\text{execution price} - \text{median of both asking and bid quotes}|}{\text{median of both quotes}} \quad (6)$$

We measure the changes in these indicators before and after the announcement of the introduction of the long-term shareholder perks program. Specifically, the period from 140 business days before the announcement to 21 business days prior to the pre-announcement period is considered the pre-announcement period and the period from 21 business days following the announcement to 140 days after the announcement is considered the post-announcement period. The differences in the spread rates are adjusted by the differences in the control sample to verify whether significant differences are observed.

Changes in medium- and long-term liquidity are measured using Amihud's (2002) illiquidity, which is an index (Eq (7)) that divides the daily stock return (absolute value) by the trading value of the stock on the same day, representing the percentage change in the stock price for a ¥1 trade. A higher number indicates a greater impact on the volume of stock orders on the price of a stock, which is interpreted as lower liquidity.

$$Illiquidity_{i,y} = \frac{1}{D_{i,y}} \sum\nolimits_{t=1}^{Diy} \frac{|R_{iyd}|}{VOLD_{iyd}} * 1,000,000 \quad (7)$$

where $R_{iyd}$ is the return on stock $i$ on day $d$ of year $y$ and $VOLD_{iyd}$ is the value of daily volume in yen. $D$ is the number of days for which data are available for stock $i$ in year $y$.

In this study, the starting point (year-1) is the period from one and a half years before the announcement of the introduction of the long-term shareholder perks program (day-360) to day-120, and the averages for the period from day+120 to day+360 is considered year+1, from day+361 to day+600 is considered year+2, and from day+601 to day840 is considered year+3.

The illiquidity of each period is calculated and the differences from year-1 to year+1, year +2, and year+3 are adjusted by the differences in the control sample to test whether significant differences are found.

## Sources of stock price effects

To determine whether the number of individual shareholders and stock liquidity are the source of the CAR at announcement, we conduct a cross-sectional regression analysis with the CAR of each sample firm as the explained variable and the indicators for individual shareholders and stock liquidity as the explanatory variables. Note that each explanatory variable is measured both before and after the announcement, and is considered a future event at the time of the announcement. Therefore, a significant correlation between CAR and each explanatory variable suggests that expectations regarding changes in these variables were the source of the CAR. The specific proxy variables and their underlying hypotheses are presented in Table 3.

Table 4 presents the basic statistics for the samples used in this study.

## Results

### Announcement effect of introducing a long-term shareholder perks

Fig 4 illustrates the CAR trend. The day of the announcement of the introduction of the long-term shareholder perks program is set as day0, and the ARs for 10 days before and after the announcement are accumulated. The results for the entire sample (dotted line) show that the CAR increases around day0 and remains at around 2% on day 10. A similar trend is observed when the sample is divided into favorable and unfavorable changes; however, the CAR for

**Table 3. Hypotheses and proxy variables.**

| | Hypothesis | Proxy Variables for Hypotheses | Mark |
|---|---|---|---|
| H1 | Announcement of a favorable long-term shareholder perk program generates a positive stock return. | CAR (0, +1) | Positive |
| H2-1 | Stock return is positively correlated with an increase in the ratio of individual shareholders. | DID_Ratio of individual shareholders | Positive |
| | | DID_Individual Shareholding Ratio | Positive |
| H2-2 | Stock return is positively correlated with stock liquidity | Quote/Execution Spread | Negative |
| | | DID_illiquidity | Negative |

favorable changes appears to be higher. These results indicate that announcements of the introduction of long-term shareholder perks may have a positive stock price effect.

Table 5 presents the mean values of ARs for the 21 days before and after the announcement date. The results of the t-test and the Wilcoxon rank sum test include the null hypothesis of mean and median AR = 0 for each. Examining the total sample, the mean AR was 0.35% for day 0, 0.70% for day +1, 0.24% for day +2, and 0.25% for day +3, each of which was statistically significant. The median ARs of 0, +1, and +2 days were also significantly positive. Since we excluded samples with simultaneous public events such as earnings announcements, these positive ARs suggest that the stock market specifically reacted to the introduction of the long-term shareholder perks program.

When the sample is dichotomized into favorable and unfavorable change, the ARs for day0, day+1, day+2, and day+3 are significantly positive in the favorable change sample. In contrast, the ARs for the unfavorable change sample, while positive, are not significant for mean or median.

Table 6 Panel A shows the results of the t-test and the Wilcoxon rank sum test with 0 as the null hypothesis for the mean and median of CAR from day0 to day+1. The mean and median CAR (day0→+1) for the entire sample were 1.04% and 0.77%, respectively, both of which are significantly positive at the 1% level. At first glance, these results suggest that long-term shareholder perks have a positive effect on stock prices in general; however, a different result emerges when the sample is divided.

Table 6 Panel B presents the results of the same analysis as Panel A, dividing the sample into favorable and unfavorable changes. The mean and median CAR (day0→+1) for the sample of favorable changes are 1.26% and 0.84%, respectively, which are both significant at the 1% level. In contrast, the mean and median values of the CAR for the unfavorable change sample are positive but not statistically significant.

**Table 4. Basic statistics of the samples.**

| | Favorable change | | | | | Unfavorable change | | | | |
|---|---|---|---|---|---|---|---|---|---|---|
| | n | Min | Mean | Median | Max | n | Min | Mean | Median | Max |
| CAR (0,−1) | 128 | −9.82 | 1.26 | 0.84 | 16.58 | 75 | −14.59 | 0.68 | 0.76 | 13.94 |
| Individual shareholding ratio (y-1) | 128 | 5.50 | 39.15 | 36.92 | 94.56 | 75 | 5.17 | 41.21 | 38.95 | 88.66 |
| Ratio of individual shareholdings (y0) | 127 | 0.81 | 38.52 | 36.65 | 96.38 | 75 | 3.26 | 42.02 | 41.52 | 88.84 |
| Ratio of individual shareholders (y-1) | 128 | 69.88 | 95.83 | 97.10 | 99.41 | 75 | 89.41 | 97.50 | 98.04 | 99.55 |
| Ratio of individual shareholders (y0) | 127 | 70.01 | 96.09 | 97.23 | 99.45 | 75 | 89.72 | 97.44 | 97.96 | 99.54 |
| ROA | 119 | −4.69 | 7.23 | 5.28 | 29.75 | 68 | −1.44 | 7.35 | 6.04 | 37.25 |
| PBR | 126 | 0.30 | 2.10 | 1.19 | 44.93 | 75 | 0.20 | 2.08 | 1.23 | 19.58 |
| illiquidity (y-1) | 123 | 0.0002 | 0.127 | 0.023 | 2.311 | 71 | 0.00004 | 0.082 | 0.024 | 0.754 |
| Quoted spread (immediately before announcement) | 128 | 0.09 | 0.46 | 0.34 | 3.15 | 75 | 0.09 | 0.47 | 0.35 | 2.71 |
| Execution spread (just before announcement) | 128 | 0.03 | 0.16 | 0.10 | 1.52 | 75 | 0.02 | 0.16 | 0.11 | 0.99 |

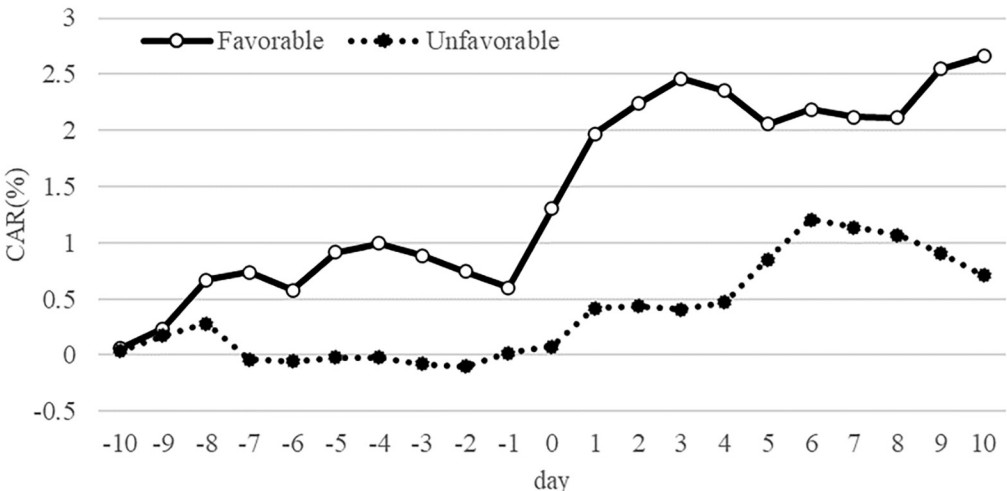

**Fig 4. Changes in cumulative abnormal return.** Note: The horizontal axis is the days prior to and following the date of the long-term shareholder perks program announcement, which is day0.

**Table 5. Announcement effect of introducing long-term shareholder perks.**

| Day | All samples | | | | | Favorable change (n = 128) | | | | | Unfavorable change (n = 75) | | | | |
|---|---|---|---|---|---|---|---|---|---|---|---|---|---|---|---|
| | AR | | | | | AR | | | | | AR | | | | |
| | Mean | t-value | | Median | Test statistic | | Mean | t-value | | Median | Test statistic | | Mean | t-value | | Median | Test statistic |
| −10 | 0.14 | 1.25 | | 0.03 | 968 | | 0.26 | 1.75 | * | 0.04 | 628 | | −0.05 | −0.29 | | 0.03 | −8 |
| −9 | 0.15 | 0.91 | | 0.11 | 572 | | 0.12 | 0.59 | | 0.13 | 327 | | 0.19 | 0.73 | | 0.06 | 32 |
| −8 | 0.23 | 1.34 | | 0.05 | 920 | | 0.37 | 1.47 | | 0.11 | 618 | | 0.01 | 0.03 | | −0.08 | −16 |
| −7 | 0.02 | 0.11 | | −0.02 | −123 | | 0.14 | 0.75 | | 0.03 | 245 | | −0.19 | −1.00 | | −0.10 | −202 |
| −6 | −0.10 | −0.79 | | 0.02 | −460 | | −0.11 | −0.66 | | 0.12 | −84 | | −0.09 | −0.44 | | −0.19 | −128 |
| −5 | 0.11 | 0.68 | | −0.05 | −234 | | 0.16 | 0.73 | | −0.10 | −133 | | 0.01 | 0.05 | | −0.02 | −13 |
| −4 | 0.11 | 0.84 | | 0.09 | 670 | | 0.13 | 0.75 | | 0.03 | 237 | | 0.07 | 0.37 | | 0.19 | 108 |
| −3 | 0.00 | 0.00 | | −0.02 | 24 | | 0.05 | 0.30 | | 0.02 | 195 | | −0.09 | −0.66 | | −0.15 | −117 |
| −2 | −0.02 | −0.17 | | −0.06 | −136 | | −0.03 | −0.22 | | −0.05 | 14 | | 0.00 | 0.01 | | −0.06 | −65 |
| −1 | −0.08 | −0.65 | | −0.09 | −113 | | −0.21 | −1.51 | | −0.17 | −447 | | 0.15 | 0.66 | | 0.01 | 214 |
| 0 | 0.35 | 2.00 | ** | 0.13 | 1,480 * | | 0.54 | 3.04 | *** | 0.18 | 952 ** | | 0.02 | 0.05 | | −0.06 | 8 |
| 1 | 0.70 | 3.04 | *** | 0.55 | 2,760 *** | | 0.72 | 2.65 | *** | 0.59 | 1,365 *** | | 0.66 | 1.58 | | 0.34 | 192 |
| 2 | 0.24 | 1.80 | * | 0.18 | 1,444 * | | 0.29 | 1.84 | * | 0.30 | 880 ** | | 0.15 | 0.63 | | −0.02 | 15 |
| 3 | 0.25 | 1.91 | * | 0.11 | 1,332 | | 0.33 | 1.90 | * | 0.15 | 813 * | | 0.11 | 0.57 | | −0.03 | −6 |
| 4 | −0.02 | −0.15 | | −0.13 | −521 | | −0.11 | −0.67 | | −0.11 | −362 | | 0.13 | 0.57 | | −0.14 | 17 |
| 5 | 0.01 | 0.05 | | −0.07 | 119 | | −0.23 | −1.31 | | −0.17 | −578 | | 0.40 | 1.86 | * | 0.39 | 364 * |
| 6 | 0.04 | 0.27 | | 0.08 | 62 | | −0.00 | −0.01 | | −0.18 | −317 | | 0.12 | 0.44 | | 0.26 | 203 |
| 7 | −0.16 | −1.18 | | −0.10 | −1418 * | | −0.07 | −0.39 | | −0.08 | −314 | | −0.31 | −1.64 | | −0.21 | −337 * |
| 8 | 0.12 | 0.76 | | 0.11 | 536 | | 0.10 | 0.49 | | 0.10 | 182 | | 0.15 | 0.63 | | 0.11 | 113 |
| 9 | 0.27 | 2.38 | ** | 0.28 | 1,834 ** | | 0.44 | 2.97 | *** | 0.32 | 1,118 *** | | −0.03 | −0.16 | | −0.03 | 18 |
| 10 | −0.12 | −0.82 | | 0.01 | −957 | | −0.15 | −0.70 | | −0.01 | −446 | | −0.07 | −0.42 | | 0.08 | −86 |

Day is the date of the announcement of the introduction of the long-term shareholder perks program as base date 0.

A t-test and a Wilcoxon rank sum test are conducted using mean = 0 and median = 0 for AR as the null hypotheses, respectively.

***, **, and * indicate statistical significance at 1%, 5%, and 10% levels, respectively.

**Table 6. Test results for Cumulative Abnormal Return (CAR).**

| Panel A | | | | | | | | | | | | | | |
|---|---|---|---|---|---|---|---|---|---|---|---|---|---|---|
| | | All samples | | | | | | | | | | | | |
| | n | Mean | t-value | | Median | Test statistic | | | | | | | | |
| CAR(day0→+1) | 203 | 1.04 | 3.66 | *** | 0.77 | 3133 | *** | | | | | | | |
| Panel B | | | | | | | | | | | | | | |
| | | Favorable change | | | | | | | Unfavorable change | | | | | |
| | n | Mean | t-value | | Median | Test statistic | | n | Mean | t-value | | Median | Test statistic | |
| CAR(day0→+1) | 128 | 1.26 | 3.79 | *** | 0.84 | 1484 | *** | 75 | 0.68 | 1.29 | | 0.75 | 297 | |

*** indicates statistical significance at the 1% level.

The findings indicate that the introduction of long-term shareholder perks has a positive stock price effect in the favorable change case, whereas the introduction of long-term shareholder perks did not have a positive effect on stock prices in the case of an unfavorable change for new individual investors or for those who intended to hold the shares for a short period of time; however, negative stock price effects are also not observed in the case of an unfavorable change.

## Changes in shareholder base

The primary purpose of publicly traded companies introducing shareholder perk programs is to attract individual investors. Previous studies have shown that the number of individual investors increases significantly in companies that introduce new shareholder perk programs [2,18]. Does the introduction of long-term shareholder perks increase the number of individual investors?

Table 7 presents the changes in the ratio of individual shareholders (number of individual shareholders divided by the total number of shareholders) and the ratio of individual shareholder holdings (number of individual shareholder holdings divided by the number of shares outstanding) before and after the introduction of the long-term shareholder perks program (the period immediately before and after the introduction).

The top panel of the table shows the before and after of the sample itself, without calculating the difference from the control sample. The average of all samples shows that the share of individual shareholders increased by 0.15 percentage points after the changes. When the sample is

**Table 7. Changes in the ratio of individual shareholders and individual shareholdings.**

| | | Individual Shareholder Ratio | | | | | | Individual Shareholding Ratio | | | | | |
|---|---|---|---|---|---|---|---|---|---|---|---|---|---|
| | n | Mean of Difference | t-value | | Median of Difference | Test statistic | | Mean of Difference | t-value | | Median of Difference | Test statistic | |
| No adjustment | | | | | | | | | | | | | |
| All samples | 202 | 0.15 | 2.42 | ** | 0.04 | 1,605 | * | 0.13 | 0.16 | | 0.04 | 138 | |
| Favorable change | 127 | 0.27 | 3.09 | *** | 0.06 | 1,063 | *** | −0.27 | −0.22 | | 0.05 | −90 | |
| Unfavorable change | 75 | −0.06 | −0.93 | | −0.01 | -56 | | 0.81 | 1.15 | | 0.03 | 121 | |
| DID | | | | | | | | | | | | | |
| All samples | 202 | 0.08 | 1.22 | | 0.01 | 269 | | 0.30 | 0.34 | | 0.24 | 714 | |
| Favorable change | 127 | 0.23 | 2.63 | *** | 0.08 | 790 | * | 0.04 | 0.03 | | 0.28 | 325 | |
| Unfavorable change | 75 | −0.17 | −1.78 | * | −0.11 | −378 | ** | 0.74 | 0.85 | | 0.07 | 97 | |

***, **, and * indicate statistical significance at 1%, 5%, and 10% levels, respectively.

divided into favorable and unfavorable changes, an increase of 0.27 percentage points results for favorable changes, which is significant at the 1% level. The lower panel of Table 7 shows the DID results adjusted for change in the control sample that are selected using PSM. An increase of 0.23 percentage points is observed in the sample of favorable changes, which is significant at the 1% level. The median test is also significantly positive at the 10% level. In the case of favorable changes, long-term shareholder perks appear to have the effect of increasing the number of individual shareholders or the shareholder base. When the shareholder base expands, investors with diverse opinions gather, and this collective knowledge lowers the transaction costs associated with information asymmetry, which then lowers the overall cost of capital and has a positive impact on stock prices [20]. We interpret the positive CAR as indicating that investors who encountered the announcement of long-term shareholder perks anticipated a subsequent decrease in the cost of capital due to an expected increase in the number of individual shareholders.

Conversely, the number of individual shareholders decreases in the unfavorable change sample, although no significant positive stock price effect was observed. The DID results show that the mean and median are −0.17 percentage points and −0.11 percentage points, significant at 10% and 5% level, respectively. Our interpretation of this phenomenon is as follows.

First, an unfavorable change in long-term shareholder perks is effective in retaining existing individual shareholders. However, it must be noted that not all individual shareholders invest with the primary goal of obtaining these perks. While these perks may be attractive, various other factors can influence shareholders to sell their shares. In the event of a favorable change, an increase in new individual shareholders is expected, which could compensate for any decrease in the existing shareholder base. Conversely, an unfavorable change is less attractive to new individual shareholders; therefore, the number of individual shareholders is reduced overall.

Second, individual investors may assess whether a change is favorable or unfavorable at the same time the long-term shareholder perk is announced. In Japan, information is actively shared through social networking services, and notably unfavorable changes may be reported as "sad news." Such information sharing could serve as a signal for potential reduction in the number of new individual shareholders.

Third, although unfavorable changes in long-term shareholder perks lead to a decrease in the number of new individual shareholders and cause a decrease in the overall number of individual shareholders, they do not have a negative impact on the share price, at least not because of the benefits favoring existing individual shareholders.

Individual shareholdings does not change significantly in either the unadjusted analysis or the DID for both the favorable and unfavorable change samples. Although the number of individual shareholders increases with a favorable change, it does not significantly increase the percentage of individual shareholdings. Unlike institutional investors, individual shareholders have limited purchasing power. In addition, the yield on shareholder perks tends to be highest at a minimum lot of 100 shares, and new individual shareholders prefer to invest small amounts at the minimum lot; therefore, the number of shares held by individual shareholders does not increase as much as the number of shareholders.

## Changes in equity liquidity

The introduction of new shareholder perk programs increases stock liquidity [18]. In contrast, some scholars have suggested that long-term shareholder perk programs have a negative impact on liquidity by immobilizing individual shareholders [6]. These previous studies introduced conceptual considerations, and the quantitative empirical results were unclear.

**Table 8. Change in equity liquidity (short-term).**

| | n | Difference in Quote Spread | | Difference of Execution Spread | | DID of Quote Spread | | DID of Execution Spread | | |
| --- | --- | --- | --- | --- | --- | --- | --- | --- | --- | --- |
| | | Mean | Median | Mean | Median | Mean | Median | Mean | Median | |
| All samples | 203 | −0.007 | −0.004 | −0.004 | −0.002 | −0.026 | −0.006 | −0.014 | −0.009 | * |
| Favorable change | 128 | −0.021 | −0.009 | −0.009 | −0.002 | −0.049 | −0.008 | −0.021 * | −0.009 | ** |
| Unfavorable change | 75 | 0.026 | 0.012 | 0.007 | −0.000 | 0.024 | 0.011 | 0.000 | −0.007 | |

** and * indicate statistical significance at 5% and 10% levels, respectively.

Therefore, we measure stock liquidity before and after the introduction of long-term shareholder perks.

Table 8 presents the quote and execution spread from 120 business days before to 21 business days before the announcement of the long-term shareholder perk (day-120 to -21) and from day+21 to +120. Each represents the amount of change before and after the announcement. The left-hand side presents the results of the sample only. For the total sample, the quoted spread is reduced by −0.007 percentage points on average and by −0.004 percentage points at the median. The average and median execution spreads are also reduced by −0.004 percentage points and −0.002 percentage points, respectively. Splitting the sample into favorable and unfavorable change categories indicates that the spread reduction is more pronounced for favorable changes; however, neither the t-test for the mean nor the Wilcoxson signed rank test for the median is significant.

The right-hand side of Table 8 shows the DID results. For favorable change, the quote spread is −0.049 percentage points lower on average and −0.008 percentage points lower at the median. The execution spread is significant at 10% and 5% levels, with respective mean and median decreases of −0.021 percentage points and −0.009 percentage points.

Long-term shareholder perk plans are found to produce additional spread reductions (improved liquidity) in the case of favorable changes. We interpret the liquidity improvement effect of a larger shareholder base as outweighing the negative effect of an increase in long term individual shareholders. In contrast, in the case of an unfavorable change, the spread does not change before adjustment or DID. Equity liquidity was not worsened by the unfavorable change.

Will the improvement in equity liquidity observed in Table 8 be sustained in the long run, or will liquidity deteriorate, as Yasutake and Nagata (2022) contended, due to the immobilization of individual shareholders? Table 9 presents the DID results for Amihud's (2002) illiquidity [6].

The starting point of the analysis is year-1, which is 360 business days (1.5 years ago, day-360) to 120 business days (6 months ago, day-120) from the announcement date of the long-

**Table 9. Changes in equity liquidity (long-term).**

| | n | DID_Iliquidity (y-1→y+1) | | | | n | DID_Iliquidity (y-1→y+2) | | | |
| --- | --- | --- | --- | --- | --- | --- | --- | --- | --- | --- |
| | | Mean | t-value | Median | Test statistic | | Mean | t-value | Median | Test statistic |
| All samples | 194 | −0.03 | −0.03 | 0.0011 | 1,084 | 192 | -0.026 | -1.044 | 0.0005 | 720 |
| Favorable change | 123 | −0.05 | −1.32 | 0.0008 | 115 | 121 | -0.03 | -0.818 | 0.0005 | 188 |
| Unfavorable change | 71 | 0.005 | 0.1766 | 0.0047 | 361 ** | 71 | -0.021 | -0.707 | 0.0006 | 177 |

Illiquidity is the percentage change in the share price (in percentage points) relative to the trading value of 1 million yen

** indicates statistical significance at the 5% level.

term shareholder perk. The endpoints of the analysis are year+1, from 121 business days to 360 business days later (day+121 to +360), and year+2, from 361 business days to 600 business days later (day+361 to +600).

No statistically significant change is observed in Amihud's (2002) illiquidity from year-1 to year+1 and year-1 to year+2 in the favorable change sample. We determine the favorable change does not lead to the worsening liquidity that previous studies have warned of. Conversely, the change was positive (i.e., worsening liquidity) for the unfavorable change sample from year-1 to year+1, and the median is significant at the 5% level. Unfavorable changes are changes that reduce investors' incentives to trade in the short term. Yasutake and Nagata (2022) demonstrated a positive correlation between the introduction of long-term perks and increased illiquidity (worsened liquidity). However, their analysis did not differentiate between favorable and unfavorable changes within their sample. Our results suggest that the positive correlation observed by Yasutake and Nagata (2022) was predominantly influenced by the sample representing unfavorable changes.

Cross trading (bridge selling) was introduced in Nose et al.(2021), referring to a transaction in which the purchase of cash shares is combined with the margin sale on the last day of the shareholder perk rights period to obtain the perk without incurring the risk of price fluctuation [3]. Cross trading is an extremely short-term transaction, but it provides equity liquidity; however, in the event of unfavorable change, investors do not receive shareholder perk in cross trades. We contend that the deterioration of liquidity from unfavorable change is due to a decrease in the number of short-term individual investors who are interested in trading, such as cross traders.

## Sources of CAR

The introduction of a favorable change to the long-term shareholder perk program yields a significant positive abnormal return. The next question is what is the source of the returns? To examine this question, we conduct a cross-sectional regression analysis using the CAR (CAR 0,+1) between the announcement date and the following day for each sample firm as the explained variable, and the favorable change dummy, DID for the ratio of individual shareholders, and DID for the ratio of individual shareholdings as explanatory variables. Table 10 presents the results. In Model 1, the favorable change dummy is not significantly correlated with the CAR. It seems that simply a favorable change does not generate additional returns. On the other hand, in Model 2, the individual shareholder ratio DID show a significant positive correlation with the CAR at the 5% level. As shown in Table 10, the long-term shareholder perk is a measure to raise the number of individual shareholders through the introduction of a new shareholder perk program. The introduction of the long-term shareholder perk has a positive impact on stock prices due to the market's expectation of an expansion in the shareholder base. Model 3 examines the DID of individual investor shareholdings, which is not significantly correlated with the CAR.

Table 11 presents the results of the hypotheses tested. First, announcements of long-term shareholder perks programs had a positive stock price effect in the favorable change sample; thus, H1 is supported for the favorable change sample. Second, the number of individual shareholders increased before and after the introduction of the program for favorable change. Moreover, the CAR is statistically significant and positively correlated with the increase in the number of individual shareholders. This suggests that the expected increase in individual shareholders is a source of abnormal returns, and H2-1 is supported for the favorable change. The shareholding ratio of individual shareholders was uncorrelated. Third, in the favorable change sample, the stock price spread narrowed before and after the introduction of the

**Table 10. Results of cross-sectional regression analysis.**

| | Explained Variables: CAR (0, −1) | | | | | |
| --- | --- | --- | --- | --- | --- | --- |
| | **Model 1** | | **Model 2** | | **Model 3** | |
| Favorable Change Dummy | 0.742 | | | | | |
| | (0.635) | | | | | |
| Ratio of individual shareholders y-1 | | | −0.058 | | | |
| | | | (0.115) | | | |
| DID individual shareholder ratio | | | 0.663 | ** | | |
| | | | (0.326) | | | |
| Individual shareholding ratio y-1 | | | | | −0.021 | |
| | | | | | (0.019) | |
| DID individual shareholding ratio | | | | | −0.006 | |
| | | | | | (0.026) | |
| ln Total Assets | −0.475 | *** | −0.469 | ** | −0.611 | *** |
| | (0.182) | | (0.184) | | (0.234) | |
| ROA | −0.002 | | −0.002 | | −0.007 | |
| | (0.068) | | (0.067) | | (0.070) | |
| PBR | −0.119 | | −0.084 | | −0.152 | |
| | (0.172) | | (0.171) | | (0.171) | |
| Intercept | 6.087 | *** | 11.969 | | 9.007 | *** |
| | (2.170) | | (11.828) | | (3.321) | |
| Adjusted R-square | 0.023 | | 0.038 | | 0.017 | |
| n | 187 | | 187 | | 187 | |

Figures in parentheses represent standard errors

***, **, and * indicate statistical significance at the 1%, 5%, and 10% level, respectively.

program. In contrast, long-term stock liquidity worsened with the unfavorable change, but did not change with the favorable change; thus, H2-2 is validated for the favorable change.

## Conclusion

This study examines the impact and source of announcements introducing long-term shareholder perks on stock prices of Japanese listed companies in the 11-year period from 2010 to the end of March 2021. We produce more precise analysis results by categorizing the total sample of 203 firms into 128 firms with favorable change and 75 firms with unfavorable change.

**Table 11. Results of hypotheses testing.**

| | Hypothesis | Proxy Variables for Hypotheses | Mark | Result |
| --- | --- | --- | --- | --- |
| H1 | Announcement of a favorable long-term shareholder perk program generates a positive stock return. | CAR (0, +1) | Positive | Supported |
| H2-1 | Stock return is positively correlated with an increase in the ratio of individual shareholders. | DID_Ratio of individual shareholders | Positive | Supported |
| | | DID_Individual Shareholding Ratio | Positive | Unsupported |
| H2-2 | Stock return is positively correlated with stock liquidity | Quote/Execution Spread | Negative | Supported |
| | | DID_illiquidity | Negative | Indirectly supported (illiquidity worsened in unfavorable change sample) |

First, an event study confirms the short-term impact on stock prices. As a result, we reveal a significant positive stock price effect in the favorable change sample. We next examine changes in the number of shareholders before and after the introduction of the long-term shareholder program, uncovering an increasing trend for favorable changes and a decreasing trend for unfavorable changes. In a multiple regression analysis using CAR as the explained variable, CAR was significantly correlated with an increase in the number of individual shareholders after the introduction of the program.

In summary, four notable points are found in this study. First, a favorable change in the long-term shareholder perks program has a positive short-term impact on stock prices. Second, a favorable change increases the number of individual shareholders after its introduction. Third, there is a positive correlation between the stock price effect and the rate of increase in the number of individual shareholders, indicating that the source of the stock price effect is investors' expectation that the long-term shareholder perk will further increase the number of shareholders. Fourth, unfavorable changes tend to have a negative impact on the number of individual shareholders and stock liquidity.

Finally, by categorizing the long-term shareholder perks program into favorable and unfavorable changes, this study conducts a more comprehensive investigation into the effects of long-term shareholder perks programs. However, shareholder perk programs significantly differ from other forms of payouts due to the wide variety of items offered. Dividends represent just one type of benefit. Each company presents its unique items, such as branded products, shopping coupons, and regional local specialties from the company's location. Thus, a detailed analysis examining long-term shareholder perks by item type is the next crucial step. The yield on shareholder perks also warrants attention. Unlike dividends, the yield on shareholder perks cannot be straightforwardly calculated; therefore, developing a highly precise method for calculating a 'yield on perks' is essential. Previous studies have indicated that the introduction of the first shareholder perks program leads to a reduction in the cost of equity capital. While not presented in this paper, we examined the correlation between the cost of equity capital (as per the CAPM model) and CARs in our samples. However, we were unable to draw any conclusions, either positive or negative, perhaps due to the limited size of the sample. Hence, these additional tests will be the focus of future research.

## Supporting information

**S1 Dataset.**
(XLSX)

**S1 File. Excel files for figures and tables.**
(XLSX)

## Author Contributions

**Conceptualization:** Yoshiaki Nose.

**Data curation:** Yao Gao.

**Funding acquisition:** Yoshiaki Nose.

**Investigation:** Yao Gao.

**Project administration:** Yoshiaki Nose.

**Resources:** Yao Gao.

**Software:** Yao Gao.

**Supervision:** Yoshiaki Nose.

**Validation:** Yoshiaki Nose.

**Visualization:** Yao Gao.

**Writing – original draft:** Yao Gao.

**Writing – review & editing:** Yoshiaki Nose.

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
