## [Decision Letter · Decision Letter 0]

4 Dec 2023

PONE-D-23-33459Long-term shareholder perks and stock price reactionPLOS ONE

Dear Dr. Nose,

Thank you for submitting your manuscript to PLOS ONE. After careful consideration, we feel that it has merit but does not fully meet PLOS ONE’s publication criteria as it currently stands. Therefore, we invite you to submit a revised version of the manuscript that addresses the points raised during the review process.

We look forward to receiving your revised manuscript.

Kind regards,

Junhuan Zhang, PhD

Academic Editor

PLOS ONE

Journal Requirements:

   "Yoshiaki Nose thanks Japan Society for the Promotion of Science (JSPS) Grants-in-Aid for Scientific Research (22K01566) for financial support. The funder had no role in study design, data collection and analysis, decision to publish, or preparation of the manuscript"

  "Yoshiaki Nose thanks Japan Society for the Promotion of Science (JSPS) Grants-in-Aid for Scientific Research (22K01566) for financial support. The funder had no role in study design, data collection and analysis, decision to publish, or preparation of the manuscript."

Reviewers' comments:

Reviewer's Responses to Questions

**Comments to the Author**

1. Is the manuscript technically sound, and do the data support the conclusions?

Reviewer #1: Yes

Reviewer #2: Partly

2. Has the statistical analysis been performed appropriately and rigorously? 

Reviewer #1: Yes

Reviewer #2: No

3. Have the authors made all data underlying the findings in their manuscript fully available?

Reviewer #1: Yes

Reviewer #2: Yes

4. Is the manuscript presented in an intelligible fashion and written in standard English?

Reviewer #1: No

Reviewer #2: No

5. Review Comments to the Author

Reviewer #1: This paper examines the effect of long-term shareholder perks on the stock prices of Japanese companies and the factors influencing market responses. The topic is intriguing, and the methodology employed is appropriate. The study aims to contribute to related research in two significant ways. Firstly, it focuses on the impact of additional long-term perks, distinguishing itself from prior studies that primarily examine initial shareholder perks. Secondly, the study categorizes the sample into favorable and unfavorable changes.

A potential weakness of this study lies in its exclusion of relevant variables that could potentially influence stock price reactions. For example, Karpoff et al. (2021) argue that, beyond factors such as share liquidity and signaling to investors, the equity cost of capital serves as a channel through which perks contribute to firm value. Furthermore, Huang et al. (2022) demonstrate that various forms of perks, such as gift cards and gift-in-kind, also affect market reactions. Consequently, I recommend that the author consider incorporating these variables into the study or, at the very least, acknowledge the possibility that other factors may impact stock price responses.

Reviewer #2: Comments

It is interesting that authors classify an introduction of long-term shareholder perks as favorable and unfavorable changes and show different results (an announcement of an introduction of long-term shareholder perk is associated with positive cumulative abnormal return only when the announcement is considered favorable). However, I have a concern about the hypotheses building and on the analyses method.

The main hypothesis for this study seems that an introduction of long-term shareholder perks has positive impact on stock price when the introduction of the long-term shareholder perks is considered as a favorable change, through an improvement in liquidity. The results are consistent with this hypothesis, that an introduction of long term shareholder perks as favorable change is associated with a positive CAR, increases in the ratio of individual shareholders/shareholdings ratios, and an improvement in equity liquidity. However, I have a difficulty in understanding the hypotheses and the methodology summarized in Table 3 Hypotheses and Proxy Variables.

Hypotheses H2-1 states, “Stock return is positively correlated with an increase in the number of shareholders”. In Table 3, the proxy variables for Hypothesis H2-1 are DID_Ratio of individual shareholders and DID_Individual shareholding Ratio. In the Result section (from page 20), Table 7 shows the changes in these shareholders/shareholding ratios with a t-test for the mean and a Wilcoxon signed rank test for the median for these variables with the null hypothesis set to 0. In the Sources of CAR section (from page 24), Table 10 shows the results of Cross-sectional regression analyses.

I wonder the Mark “Positive” for the two proxy variables in Table 3 refers to the positive and significant increases in the means and medians for adjusted and non-adjusted shareholders/shareholding ratios from Table 7, or the positive and significant coefficient on the DID individual shareholder ratio from Table 10.

If the authors are testing if the long-term shareholder perks lead to an increase in the individual shareholders/shareholding ratios, then the hypothesis should state that “an introduction of long-term shareholder perks (not the stock price effect) is associated with an increase in the ratio (not number) of individual shareholders”.

If the authors are testing H2-1 as it is, by regressing the cumulative abnormal return around the announcement (from day 0 to day +1) on the changes in the ratios of individual shareholders and individual shareholdings from before the announcement (-1 period) to after the announcement (+1 period), I doubt the validity of this hypothesis and its estimation model.

Page 14 line 373-376 states “Note that each explanatory variable is measured before and after the announcement and is a future event at the time of the announcement. Consequently, if there is a significant correlation between the CAR and each explanatory variable, this indicates that investors expected these changes at the time of the announcement.”

Do the authors mean that if there is no significant correlation between the CAR and the explanatory variables, investors do not expect the future changes at the time of the announcement? In other words, the authors are testing if the investors correctly expect the future changes or not? My guess is that the authors like to test if the expected changes in the explanatory variables are positively correlated with the CAR, assuming that the investors can expect the changes correctly.

If investors can correctly predict the future changes in the number (or ratios) of individual shareholders/shareholdings, then the positive and significant coefficient on the individual shareholders/holdings ratio could be interpreted that a correctly predicted increase in these ratios are associated with positive cumulative abnormal returns. However, I wonder how we can ensure that investors can correctly predict the future changes in these ratios at the time of the announcement. The prediction maybe incorrect. In that case, even if an increase is expected, there should be no positive abnormal return. Similarly, even if an increase is not expected, there should be positive abnormal return if the expectation is incorrect (in fact there is an increase in the ratios). I have a difficulty in understanding the logic of this estimation model which supposedly assumes that investors can correctly predict the future changes in the shareholder base at the time of the announcement. It may be due to my ignorance of the estimation model that uses future changes in some variables as explanatory variables for Cumulative Abnormal Returns in an event study. I would appreciate if the authors can reference some other studies that uses similar model.

It may be possible that observing the positive announcement return, the number of investors increases (a reverse causality may be possible). Instead of using future change as an explanatory variable, I suggest using other variables which can help investors to predict future changes, such as the ratio of individual shareholders/shareholdings before the announcement. If a reasonable proxy for the expected change in the shareholders/shareholding ratio is used, the regression model can be suitable to test a hypothesis that “stock return is positively correlated with an increase in the expected shareholders/shareholding ratios.” I have the same concern for Hypothesis H2-2 and the regression model.

The main findings in this study are the different effect of an announcement of an introduction of the long-term shareholder perks for the case of favorable and unfavorable changes. I suggest the authors to restate the hypotheses that center on the difference between the favorable and unfavorable changes, rather than the correlation between the stock price effects and the changes in the shareholders/shareholding ratios and liquidity measures.

Below are some minor comments to be considered.

- Line 35. I suggest removing “beyond the usual perks”

- Line 58: “Second, a favorable change in long-term shareholder perks coincides with a significant rise in the number of individual shareholders before and after the introduction of the program” A change in shareholder perks policy at one point in time and a chance in the number of shareholders over time can “coincide”? I feel it’s more reasonable to say that a change (introduction of shareholder perks) can cause, lead, or results in an increase in the number of shareholders.

- Line 64: “while there is no negative impact on stock liquidity due to an increased number of fixed shareholders”. It’s not clear what is meant by ”fixed” shareholders.

- Line 43: “The second is to obtain some insight into the source of the identified stock price effect.” In the remaining part of the introduction, increase/decrease in the number of shareholders and liquidity are mentioned, but not the results from the regression analysis. By showing the changes in the shareholder base and liquidity, authors are arguing that these are the sources of the announcement return, or authors are interpreting the regression analysis in the introduction section?

- Line 78-79:” Shareholder perks are payout programs in which a company offers its own goods, services, or vouchers as special gifts to shareholders who meet certain longevity conditions [7].” Is the word “longevity” appropriate?

- Line 83: “long-term shareholder incentive program perks” in other parts, “long-term shareholder perks programs” is used.

- Line 119-120 “This is a change that is more attractive to new individual investors (favorable change)”: “More” attractive than what (or who)? Isn’t it attractive for existing shareholders, too? If you like to insist the change (announcement) is attractive or not for “new” investors, then you should include “new” (investors). For example, in line 123-125 “In the second case, the shareholder perks that a new individual shareholder receives in the first year are reduced or eliminated, which is considered to be a change that is less attractive to new individual investors (unfavorable change)”

- Line 168 Karpoff et al. (2020) should be (2021)

- Line 269: the daily excess return (ARi) should be abnormal returns (ARi)?

- Line 330-335 ”The quote spread ratio is the difference between the highest bid and lowest offer prices in the board information divided by the median of the two (Equation (5))”: In equation (5), the numerator is “last asking price-last bid price”. Are (last asking price-last bid price) the same as the (highest bid price-lowest offer)?

- Line 392,406, 411 Ars should be ARs

- Page 21, 2nd paragraph, line 2: “The average of all samples shows that the share of individual shareholders increased by 0.15% after the changes,” should be “increased by 0.15 percentage points”?

- Page 21, 2nd paragraph, Line 6: “An increase of 0.23% is observed in the sample of favorable changes, which is significant at the 1% level.” 0.23 percentage points?

- Page 21, last paragraph. “First, adverse change in long-term shareholder perks is effective in retaining existing individual shareholders but is less attractive to new individual shareholders; therefore, the number of individual shareholders is reduced overall.” If the long term shareholder perks is effective in retaining existing shareholders, why the number of individual shareholders is reduced?

- Page 21, last paragraph. “Second, at the time of the announcement of the long-term shareholder perks, individual investors in the stock market will judge whether the changes are favorable or unfavorable in terms of increasing the number of new individual shareholders.” I’m not sure why this is an interpretation of the decrease in the number of individual shareholders in the unfavorable change sample.

- Page 21, last paragraph. “Third, even if the number of new individual shareholders decreases due to an unfavorable change, the impact on the share price is negligible, since most shareholders hold only a minimum number of shares.” Again, I’m not sure why this is an interpretation of the decrease in the number of individual shareholders in the unfavorable change sample. Also, if the impact on the share price is negligible because most shareholders hold only a minimum number of shares, then an increase in the shareholder base should also be negligible.

- Page 23 second paragraph line1, the “indicative spread” should “quote spread”?

- Page 24 first paragraph: “Conversely, the change was positive (i.e., worsening liquidity) for the unfavorable change sample from year-1 to year+1, and the median is significant at the 5% level. Adverse changes are changes that reduce investors’ incentives to trade in the short term.” For the unfavorable change, the worsening liquidity is consisted with the result in the previous study (Yasutake and Nagata, 2022)?

- Page 23 second paragraph “We contend that the deterioration of liquidity from unfavorable change is due to a decrease in the number of short-term individual investors who are interested in trading, such as cross traders.” ---If possible, how about testing the decrease in cross-trading (the balance of short interest), as in Nose (2021)?

6. PLOS authors have the option to publish the peer review history of their article (what does this mean?). If published, this will include your full peer review and any attached files.

Reviewer #1: No

Reviewer #2: No

---

## [Author Response · Author response to Decision Letter 0]

18 Jan 2024

We deeply appreciate the detailed and constructive feedback provided by the reviewers. Your insights have been instrumental in enhancing the quality and clarity of our work.

We enclosed with this letter are three files:

Response to Reviewers.docx, detailing our responses to each comment from you and the two reviewers.

Revised Manuscript with Track Changes.docx, showing all modifications made to the manuscript since the previous submission.

Revised Manuscript.docx, presenting the clean version of the revised manuscript.

In our Response to Reviewers, we have endeavored to comprehensively address each point raised. We believe that we have appropriately addressed all comments and suggestions, improving the manuscript significantly.

We hope that our revisions meet the journal's standards and look forward to the manuscript's positive reevaluation. We respectfully request further peer review and are eager for any additional feedback.

Thank you for considering our work for publication in PLOS ONE, and for the opportunity to refine our manuscript based on the valuable feedback provided.

---

## [Decision Letter · Decision Letter 1]

21 Feb 2024

PONE-D-23-33459R1Long-term shareholder perks and stock price reactionPLOS ONE

Dear Dr. Nose,

Thank you for submitting your manuscript to PLOS ONE. After careful consideration, we feel that it has merit but does not fully meet PLOS ONE’s publication criteria as it currently stands. Therefore, we invite you to submit a revised version of the manuscript that addresses the points raised during the review process.

We look forward to receiving your revised manuscript.

Kind regards,

Junhuan Zhang, PhD

Academic Editor

PLOS ONE

Journal Requirements:

Reviewers' comments:

Reviewer's Responses to Questions

**Comments to the Author**

1. If the authors have adequately addressed your comments raised in a previous round of review and you feel that this manuscript is now acceptable for publication, you may indicate that here to bypass the “Comments to the Author” section, enter your conflict of interest statement in the “Confidential to Editor” section, and submit your "Accept" recommendation.

Reviewer #1: All comments have been addressed

Reviewer #2: All comments have been addressed

2. Is the manuscript technically sound, and do the data support the conclusions?

Reviewer #1: Yes

Reviewer #2: Yes

3. Has the statistical analysis been performed appropriately and rigorously? 

Reviewer #1: Yes

Reviewer #2: Yes

4. Have the authors made all data underlying the findings in their manuscript fully available?

Reviewer #1: Yes

Reviewer #2: Yes

5. Is the manuscript presented in an intelligible fashion and written in standard English?

Reviewer #1: Yes

Reviewer #2: Yes

6. Review Comments to the Author

Reviewer #1: Thank you for the response to my previous comment. I think it is better to state your preliminary analysis of CAPM-beta in a footnote.

Reviewer #2: (No Response)

7. PLOS authors have the option to publish the peer review history of their article (what does this mean?). If published, this will include your full peer review and any attached files.

Reviewer #1: No

Reviewer #2: No

---

## [Author Response · Author response to Decision Letter 1]

22 Feb 2024

We deeply appreciate the detailed and constructive feedback provided by the editorial team and the reviewers. Their insights have been instrumental in enhancing the quality and clarity of our work.

Enclosed with this letter are three files:

Response to Reviewers.docx, detailing our responses to each comment from you and the reviewer #1.

Revised Manuscript with Track Changes.docx, showing all modifications made to the manuscript since the previous submission.

Revised Manuscript.docx, presenting the clean version of the revised manuscript.

We hope that our revisions meet the journal's standards and look forward to the manuscript's positive reevaluation. We respectfully request further peer review and are eager for any additional feedback.

Thank you for considering our work for publication in PLOS ONE, and for the opportunity to refine our manuscript based on the valuable feedback provided.

---

## [Editor Report · Decision Letter 2]

26 Feb 2024

Long-term shareholder perks and stock price reaction

PONE-D-23-33459R2

Dear Dr. Nose,

We’re pleased to inform you that your manuscript has been judged scientifically suitable for publication and will be formally accepted for publication once it meets all outstanding technical requirements.

Kind regards,

Junhuan Zhang, PhD

Academic Editor

PLOS ONE
---

## [Editor Report · Acceptance letter]

24 Mar 2024

PONE-D-23-33459R2 

PLOS ONE

Dear Dr. Nose, 

I'm pleased to inform you that your manuscript has been deemed suitable for publication in PLOS ONE. Congratulations! Your manuscript is now being handed over to our production team.

Kind regards, 

on behalf of

Dr. Junhuan Zhang 

Academic Editor

PLOS ONE